**Brief Investigation**

# Dynamic regulation of murine RNA polymerase III transcription during heat shock stress

Thomas F. Nguyen, James Z.J. Kwan, Jennifer E. Mitchell, Jieying H. Cui, Sheila S. Teves 🆔 *

Department of Biochemistry and Molecular Biology, Life Sciences Institute, The University of British Columbia, Vancouver, BC V6T 1Z3, Canada

*Corresponding author: Department of Biochemistry and Molecular Biology, Life Sciences Institute, The University of British Columbia, 2350 Health Sciences Mall, Vancouver, BC, V6T 1Z3, Canada. Email: sheila.teves@ubc.ca

Cells respond to many different types of stresses by overhauling gene expression patterns, both at the transcriptional and translational levels. Under heat stress, global transcription and translation are inhibited, while the expression of chaperone proteins is preferentially favored. As the direct link between mRNA transcription and protein translation, transfer RNA (tRNA) expression is intricately regulated during the stress response. Despite extensive research into the heat shock response (HSR), the regulation of tRNA expression by RNA polymerase III (Pol III) transcription has yet to be fully elucidated in mammalian cells. Here, we examine the regulation of Pol III transcription during different stages of heat shock stress in mouse embryonic stem cells. We observe that Pol III transcription is downregulated after 30 min of heat shock, followed by an overall increase in transcription after 60 min of heat shock. This effect is more evident in tRNAs, although other Pol III gene targets are also similarly affected. Notably, we show that the downregulation at 30 min of heat shock is independent of HSF1, the master transcription factor of the HSR, but that the subsequent increase in expression at 60 min requires HSF1. Taken together, these results demonstrate an adaptive RNA Pol III response to heat stress and an intricate relationship between the canonical HSR and tRNA expression.

Keywords: heat shock; transcription; transfer RNA; mouse embryonic stem cells

## Introduction

Cells are often exposed to various environmental stressors that necessitate rapid and coordinated responses at the molecular level, including the reprogramming of gene expression (Fulda *et al.* 2010). Among the best-characterized stress responses is the heat shock (HS) response (HSR), governed by the master regulator heat shock factor 1 (HSF1) (Ritossa 1962; Lindquist 1986; Fulda *et al.* 2010). Upon HS, HSF1 is activated and induces the rapid transcription of HS protein (HSP) genes by RNA polymerase II (Pol II) (Tissiéres *et al.* 1974; Lindquist and Craig 1988; Åkerfelt *et al.* 2010; Björk and Sistonen 2010). These HSPs function as molecular chaperones, refolding misfolded proteins during heat stress to maintain proteostasis (Lindquist and Craig 1988; Richter *et al.* 2010). Simultaneously, global transcription and translation are largely inhibited to allocate cellular resources properly during heat stress (Mahat *et al.* 2016; Aprile-Garcia *et al.* 2019; Pessa *et al.* 2024).

Although most of the research on stress responses has focused on the transcriptional regulation of mRNA and stress-induced genes, the regulation of transfer RNAs (tRNAs) during stress remains less understood. As critical links between transcription and translation, the proper regulation of tRNA expression under stress conditions is essential for maintaining proteostasis. RNA polymerase III (Pol III) transcribes specific noncoding genes, including tRNAs, 5S rRNA, and U6 snRNA (Weinmann and Roeder 1974; Ullu and Weiner 1985; Krüger and Benecke 1987; Reddy *et al.* 1987). Studies in yeast and human cells show that stress

conditions, including oxidative stress and serum starvation, lead to a global reduction in tRNA abundance (Desai *et al.* 2005; Michels *et al.* 2010; Orioli *et al.* 2016; Torrent *et al.* 2018). Stress can also disrupt tRNA maturation and function, as observed with arsenite and methyl methanesulfonate in yeast (Yoluç *et al.* 2021). Intriguingly, tRNA nuclear relocalization has been proposed as a mechanism to downregulate translation during stress in both yeast and mammalian cells (Shaheen and Hopper 2005; Miyagawa *et al.* 2012). However, how Pol III regulation adapts to heat stress over time in mammalian cells, and whether the canonical HSR is involved, remains to be fully elucidated.

In this study, we focused on the regulation of Pol III-driven tRNA transcription during HS stress in mouse embryonic stem cells (mESCs). We aimed to determine how Pol III activity and tRNA transcription are modulated at different stages of HS and whether these changes are coordinated with the canonical HSR as governed by HSF1. Our findings reveal an intricate regulation of Pol III during heat stress, with tRNA transcription showing distinct temporal dynamics, and highlight the role of HSF1 in modulating these processes.

## Results

### Transcription of tRNA genes is dynamically regulated during HS

We explored how tRNA transcription adapts to heat stress over time by exposing mESCs to HS at 42°C for 30 and 60 min (HS30

and HS60) and measuring Pol III occupancy using Cleavage Under Targets and Tagmentation (CUT&Tag), a high-resolution chromatin profiling technique (Kaya-Okur *et al.* 2019), in 2 biological replicates for each condition (Supplementary Fig. 1a). These time points are consistent with previous studies delineating early/intermediate from late transcriptional response related to HS (Mahat *et al.* 2016; Vihervaara *et al.* 2021). Reads were aligned to the mouse genome, normalized using ChIP-seq-Spike-In-Free method (Jin *et al.* 2020), and replicates were merged. Gene browser tracks at specific tRNA gene loci show Pol III binding decreased 2-fold upon 30 min of HS (Fig. 1a) compared to unstressed (HS0) conditions, consistent with previous findings using different stressors in other species (Desai *et al.* 2005; Michels *et al.* 2010; Orioli *et al.* 2016). Surprisingly, Pol III binding returned back to normal under continued HS for 60 min (Fig. 1a). A similar trend was observed for all tRNA genes when we plotted the Pol III occupancy in a 400-bp window surrounding the transcription start site (TSS) for all tRNA genes (Fig. 1b).

To quantify the change in Pol III binding over time under HS, we plotted the Pol III CUT&Tag normalized read count values on tRNAs as scatter plots, comparing HS0 vs HS30, HS30 vs HS60, and HS0 vs HS60 samples (Fig. 1c). Linear regression analyses showed a ~50% decrease in HS0 vs HS30, followed by an increase of ~50% in HS30 vs HS60. The HS0 vs HS60 scatter plot confirmed the genome-wide trends. These analyses also show that both downregulation and subsequent recovery occur for all transcribed tRNAs and not a unique subset. Importantly, immunoblotting of whole-cell lysates for the RPC7 subunit of Pol III showed no changes in unstressed, HS30, and HS60 conditions, suggesting that the changes in Pol III occupancy during HS are not due to changes in Pol III protein levels (Supplementary Fig. 2a).

The CUT&Tag data showed changes in Pol III occupancy upon HS, but to directly assess Pol III activity, we performed several analyses. First, we reanalyzed previously published native elongating transcripts coupled with sequencing (NET-seq) data sets in HS30-treated mESCs compared to unstressed controls (Kwan *et al.* 2023). NET-seq captures newly transcribed RNA from elongating RNA Pols at single nucleotide resolution through extensive fractionation (Mayer *et al.* 2015). We observed a 2-fold decrease in signal at *AlaAGC* and *TyrGTA* gene loci (Supplementary Fig. 2b) and genome-wide across all tRNAs (Supplementary Fig. 2c) in HS30-treated mESCs compared to unstressed mESCs. A scatterplot comparing normalized NET-seq reads from the TSS to the transcription end site (TES) of all tRNAs in unstressed vs HS30-treated mESCs also showed a similar decrease of signal in HS30-treated cells (Supplementary Fig. 2d), corroborating the Pol III CUT&Tag data that tRNA transcription is decreased in mESCs after 30 min of HS.

Second, we reanalyzed previously published precision run-on-sequencing (PRO-seq) data sets (Vihervaara *et al.* 2021) of mouse embryonic fibroblasts (MEFs) treated with 25 and 60 min of HS (HS25 and HS60, respectively). Similar to NET-seq, PRO-seq maps the active site of RNA Pols with single base resolution, providing a direct measure of transcriptional activity (Mahat *et al.* 2016). Consistent with the CUT&Tag and NET-seq analyses, we observed a 2–3-fold decrease in PRO-seq signal at *AlaAGC* and *TyrGTA* gene loci (Fig. 1a) and genome-wide across all tRNAs (Fig. 1b; Supplementary Fig. 1a) in HS25-treated MEFs compared to unstressed MEFs. We also observed a 5–8-fold increase in *AlaAGC* and *TyrGTA* levels (Fig. 1a) and across all tRNAs genome-wide (Fig. 1b; Supplementary Fig. 1a) in HS60-treated MEFs compared to HS25 treatment and unstressed cells. These analyses show

that the dynamic HS-related changes in Pol III activity that we observed in mESCs also occur in MEFs.

Third, we performed metabolic labeling with 5′-bromo-uridine (5-BrU) to directly measure changes in tRNA expression over 30 and 60 min of HS in mESCs (Supplementary Fig. 2e). 5-BrU-labeled nascent RNA was immunoprecipitated (Imamachi *et al.* 2014) and quantified by RT-qPCR (BRI-qPCR) in unstressed, HS30-treated, and HS60-treated mESCs (Fig. 1d). In vitro transcribed 5-BrU-labeled NanoLuc Luciferase (NLuc) RNA was spiked in for normalization (England *et al.* 2016). Analyzing 2 different tRNA genes (*AlaAGC* and *TyrGTA*), we observe a decreasing trend in nascent RNA after 30 min of HS, followed by an increasing trend after 60 min (Fig. 1d). Altogether, these data indicate an adaptive and dynamic regulation of RNA Pol III transcription of tRNAs during HS.

To determine whether the observed changes in tRNA transcription are specific to heat stress, we treated mESCs with the proteasome inhibitor MG132 for 0, 120, and 240 min. MG132 has been shown to activate HSF1 and induce HS protein expression in mammalian cells (Kim *et al.* 2011). We confirmed this effect by RT-qPCR analysis of *Hspa1a*, which showed increased expression at 120 and 240 min post-treatment. However, RT-qPCR analysis of *AlaAGC* and *TyrGTA* tRNAs revealed no significant changes in tRNA transcription following MG132 treatment (Supplementary Fig. 2f). These results suggest that the observed changes in tRNA transcription are linked to the HSR beyond the direct activation of HSF1.

## HSF1 regulates the adaptive recovery of tRNA transcription during HS

As the master regulator of the HSR, HSF1 induces the Pol II-mediated transcription of HSP genes (Åkerfelt *et al.* 2010; Björk and Sistonen 2010), but how this canonical HSR feedbacks into tRNA transcription is unknown. Previously generated *Hsf1*^−/− mESCs (Price *et al.* 2023) were exposed to HS at 42°C for 30 and 60 min. We confirmed HSF1 knockout by western blot analysis (Fig. 2a), and that these cells failed to induce HSP genes, as measured by RT-qPCR of the *Hspa1a* gene (Supplementary Fig. 3a). To investigate how the chromatin occupancy of Pol III is affected upon HS treatment in the absence of HSF1, we performed Pol III CUT&Tag analysis in 2 biological replicates (Supplementary Fig. 3b) of unstressed, HS30-treated, and HS60-treated *Hsf1*^−/− mESCs, normalized using the ChIP-seq-Spike-In-Free method (Jin *et al.* 2020). Similar to wild-type cells, gene browser tracks at specific tRNA loci show that Pol III is bound abundantly in unstressed conditions and decreases 2-fold after 30 min of HS (Fig. 2b). In contrast, the observed recovery of Pol III occupancy after 60 min of HS in wild-type cells was blunted in *Hsf1*^−/− mESCs (Fig. 2b). We plotted the Pol III occupancy in a 400-bp window surrounding the TSS for all tRNA genes in unstressed, HS30, and HS60-treated *Hsf1*^−/− mESCs and observed a similar trend across all tRNA genes (Fig. 2c).

To further quantify the changes in Pol III occupancy, we generated scatterplots with normalized read count values of Pol III on tRNAs in HS0 vs HS30-treated cells, HS30-treated vs HS60-treated cells, and HS0 vs HS60-treated cells and performed linear regression analysis. We observed that most points fell below the diagonal, decreasing ~45% in the HS0 vs HS30-treated samples and increasing ~15% in HS30-treated vs HS60-treated samples (Fig. 2d). The HS0 vs HS60 scatterplot shows a ~30% overall decrease, reflecting the stunted recovery observed between these conditions. As in wild-type cells, western blot analysis of unstressed, HS30-treated, and HS60-treated *Hsf1*^−/− mESC whole-cell lysates showed comparable RPC7 levels across all conditions,

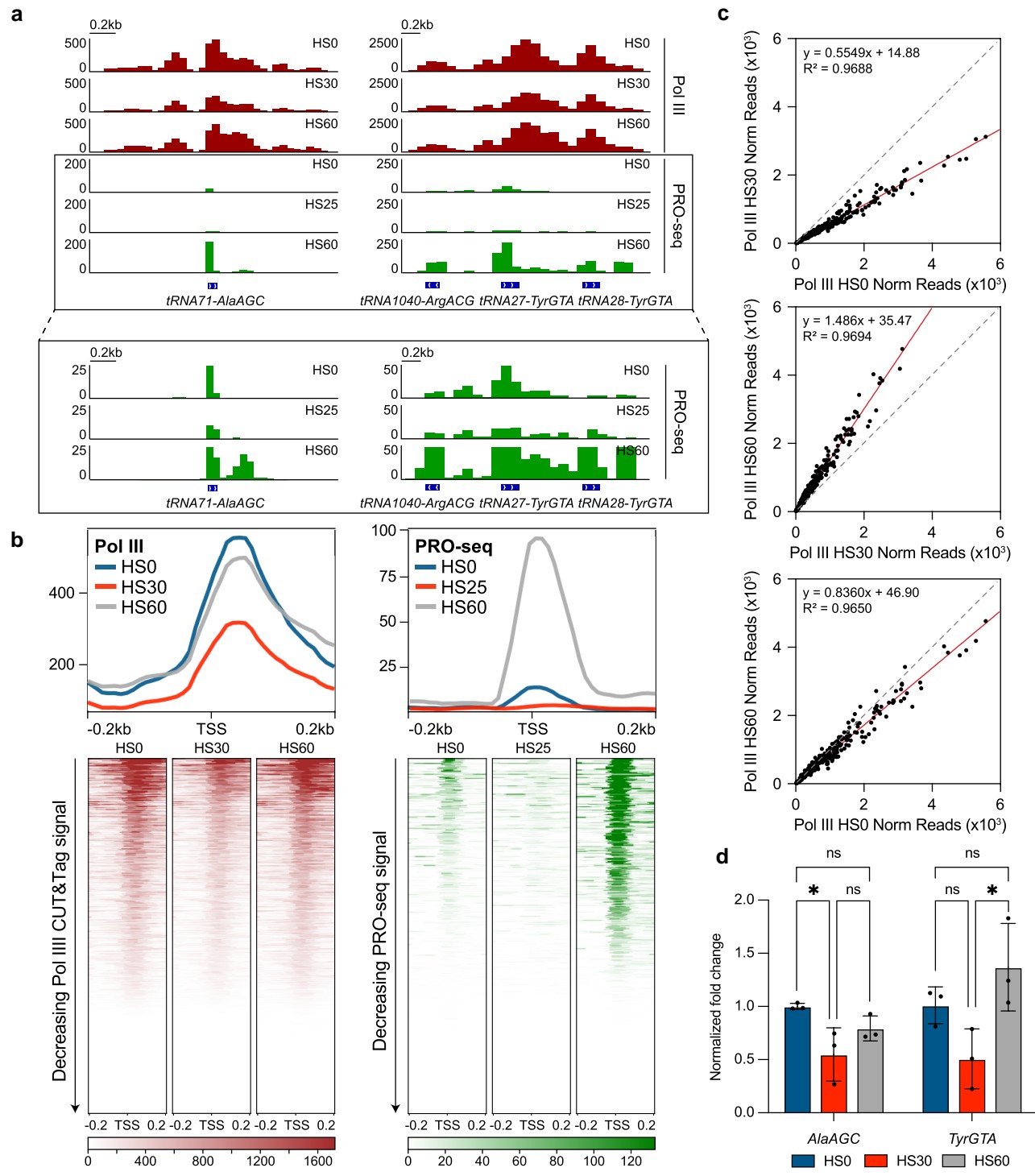

**Fig. 1.** Transcription of tRNAs is dynamically regulated during different stages of HS. a) Gene browser tracks at *tRNA71-AlaAGC* (left) and *tRNA1040-ArgACG, tRNA27-TyrGTA*, and *tRNA28-TyrGTA* (right) of Pol III CUT&Tag ($n = 2$) in unstressed mESCs (HS0) or with 30 (HS30) and 60 min (HS60) of HS treatment and PRO-seq (from Vihervaara et al. 2021, $n = 2$) in unstressed wild-type MEFs or with 25 and 60 min of HS (HS25 and HS60, respectively). PRO-seq tracks with adjusted y-axis are shown below within the boxed regions. b) Genome-wide average plots (top) and heatmaps (bottom) arranged by decreasing signal for Pol III CUT&Tag in mESCs and PRO-seq in MEFs in a 400-bp window surrounding the TSS of all tRNA genes. c) Normalized read counts of Pol III CUT&Tag signal in unstressed vs HS30-treated mESCs (top), unstressed vs HS60-treated mESCs (middle), and HS30-treated vs HS60-treated mESCs (bottom) from the TSS to the TES of all tRNA genes. d) BRI-qPCR analysis of *tRNA71-AlaAGC* and *tRNA27-TyrGTA* in HS0-, HS30-, and HS60-treated mESCs normalized to NLuc signal ($n = 3$, mean ± SD). Statistical analysis was performed using 1-way ANOVA. Ns, nonsignificant. *$P \leq 0.05$.

suggesting that HS treatment did not affect Pol III protein levels in *Hsf1*$^{-/-}$ mESCs (Supplementary Fig. 3c).

We then tested whether Pol III occupancy is consistent with activity by performing BRI-qPCR in unstressed, HS30-treated, and HS60-treated *Hsf1*$^{-/-}$ mESCs (Fig. 2e). BRI-qPCR of the tRNA target *AlaAGC* showed an average decrease in signal in HS30-treated *Hsf1*$^{-/-}$ mESCs compared to unstressed conditions and that this signal remained low in HS60-treated conditions (Fig. 2e).

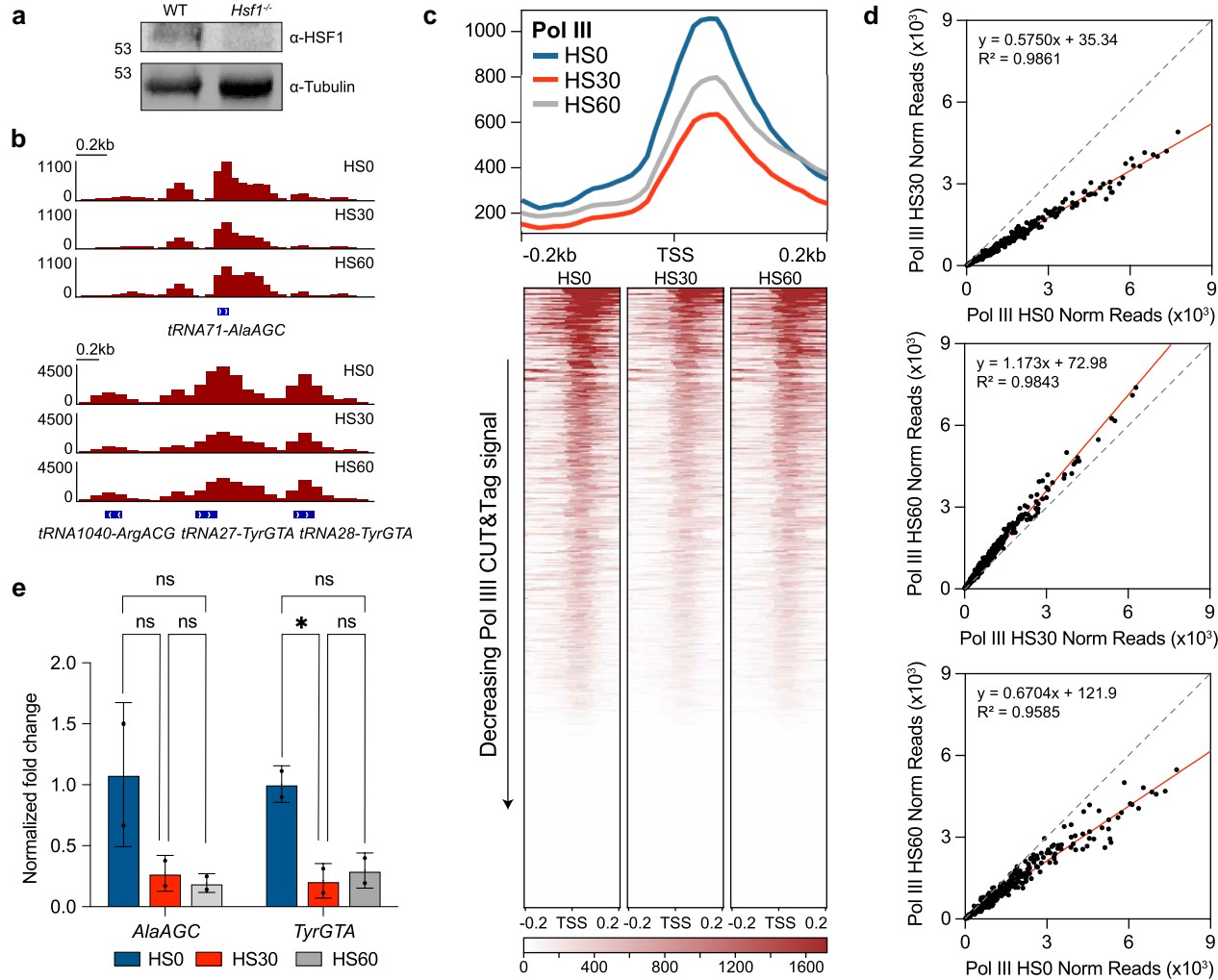

**Fig. 2.** tRNA transcription regulation is partially dependent on HSF1. a) Immunoblot analysis with α-HSF1 and α-Tubulin of WT and *Hsf1*$^{-/-}$ mESC whole-cell extracts. b) Gene browser tracks at *tRNA71-AlaAGC* (left) and *tRNA1040-ArgACG, tRNA27-TyrGTA, tRNA28-TyrGTA* (right) of Pol III CUT&Tag (maroon, $n = 2$) in unstressed *Hsf1*$^{-/-}$ mESCs (HS0) or with 30 (HS30) and 60 min (HS60) of HS treatment. c) Genome-wide average plots (top) and heatmaps (bottom) for Pol III CUT&Tag signal in a 400-bp window surrounding the TSS of all tRNA gene, arranged by decreasing signal in *Hsf1*$^{-/-}$ mESCs after HS0, HS30, and HS60 treatments. d) Normalized read counts of Pol III CUT&Tag signal in unstressed vs HS30-treated *Hsf1*$^{-/-}$ mESCs (top), unstressed vs HS60-treated *Hsf1*$^{-/-}$ mESCs (middle), and HS30-treated vs HS60-treated *Hsf1*$^{-/-}$ mESCs (bottom) from the TSS to the TES of all tRNA genes. e) BRI-qPCR analysis of *tRNA71-AlaAGC* and *tRNA27-TyrGTA* in HS0-, HS30-, and HS60-treated *Hsf1*$^{-/-}$ mESCs normalized to NLuc signal ($n = 2$, mean ± SD). Statistical analysis was performed using 1-way ANOVA. ns, nonsignificant; WT, wild type. *$P \leq 0.05$.

However, BRI-qPCR of the tRNA target *TyrGTA* showed a significant decrease in RNA levels in HS30-treated mESCs compared to unstressed conditions and that this decrease in RNA levels persists into HS60-treated conditions (Fig. 2e). Taken together, these results suggest that the downregulation of tRNAs during mid-HS is independent of HSF1, while the recovery of tRNA transcription during late HS depends on the presence of HSF1 in mESCs.

## HSF1 modulates the adaptive response of other Pol III gene classes

In addition to tRNA genes, Pol III also transcribes the 5S ribosomal RNA, a core component of the ribosome, 7S RNAs, and other small noncoding RNAs (Weinmann and Roeder 1974; Ullu and Weiner 1985; Krüger and Benecke 1987; Reddy *et al.* 1987). To investigate how these other classes of Pol III genes adapt to heat stress over time, we analyzed our Pol III CUT&Tag data at 42°C for HS30 and HS60 at the 5S, 7S1, and 7SK genes. Gene browser tracks at these gene loci show strong Pol III occupancy under unstressed conditions and a modest decrease upon 30 min of HS, although

not to the degree as seen for tRNAs (Fig. 3a). As with tRNAs, Pol III occupancy at the 5S, 7S1, and 7SK genes returned back to normal unstressed levels after 60 min of HS (Fig. 3a; Supplementary Fig. 4a). We quantified these results by averaging the normalized read counts within gene bodies for each replicate, and the bar plots for these samples confirm this trend (Fig. 3b). These results suggest that other classes of Pol III genes are dynamically regulated under HS stress, but the degree of regulation varies between these genes.

We next questioned whether HSF1 played a similar role at these other classes of Pol III genes as tRNA genes. Analyzing the Pol III CUT&Tag data in unstressed, HS30-treated, and HS60-treated *Hsf1*$^{-/-}$ mESCs at the 5S, 7S1, and 7SK genes, we observed a modest decrease in Pol III occupancy for HS30-treated *Hsf1*$^{-/-}$ cells compared to unstressed conditions (Fig. 3a; Supplementary Fig. 4a). However, the recovery in Pol III binding after 60 min of HS is blunted in the *Hsf1*$^{-/-}$ mESCs (Fig. 3a; Supplementary Fig. 4a). These trends are further quantified in bar plot format (Fig. 3c). Taken together, we conclude that these other classes of Pol III genes

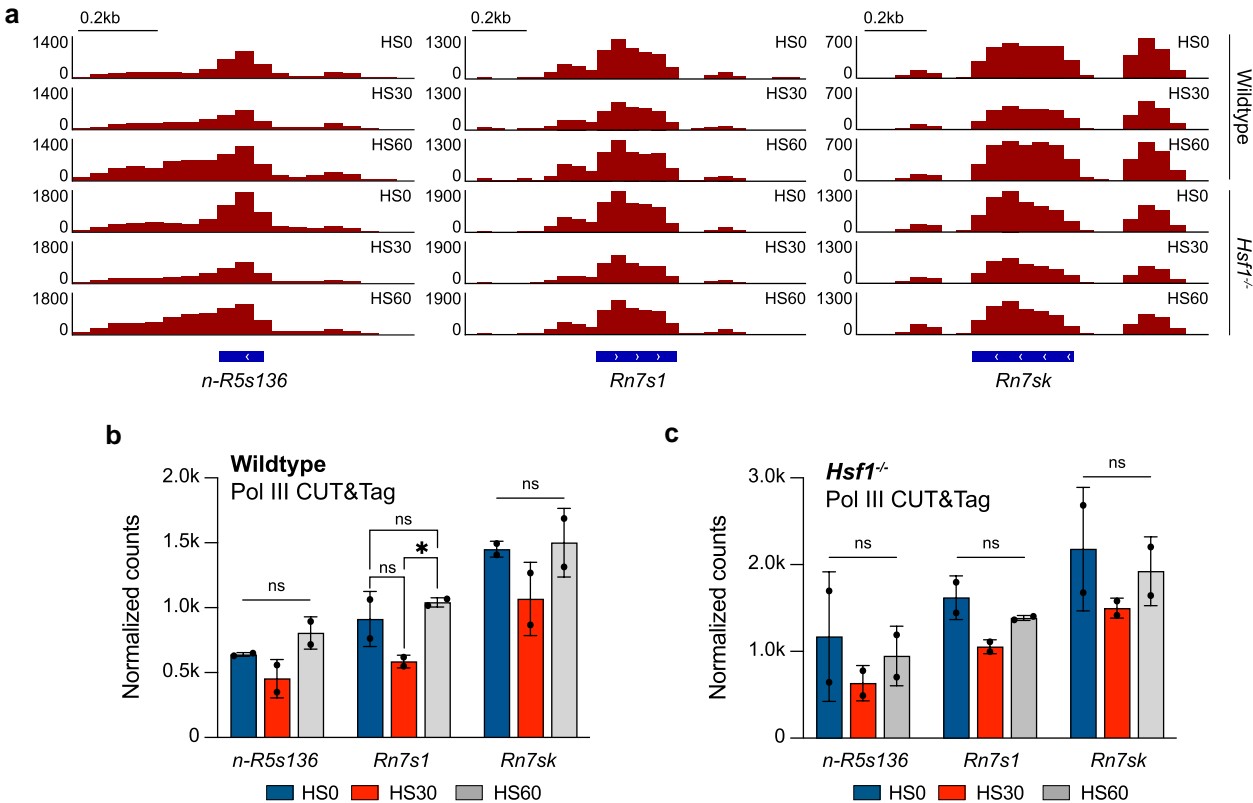

**Fig. 3.** ncRNAs transcribed by Pol III are impacted during HS. a) Gene browser tracks at *n-R5s136* (left), *Rn7s1* (middle), and *Rn7sk* (right) for reads from Pol III CUT&Tag in wild-type and *Hsf1*$^{-/-}$ mESCs with 0 (HS0), 30, (HS30), and 60 min (HS60) of HS treatment. b, c) Normalized read counts of Pol III CUT&Tag for unstressed, HS30-treated, and HS60-treated wild-type b) and *Hsf1*$^{-/-}$ c) mESCs on *n-R5s136*, *Rn7s1*, and *Rn7sk* ncRNAs. CUT&Tag counts are presented as an average of 2 biological replicates ± SD. Statistical analysis was performed using 1-way ANOVA. ns, nonsignificant. *$P \leq 0.05$.

are regulated similarly to tRNAs during stress and that HSF1 plays a conserved role in all Pol III-transcribed genes.

## HSF1 is a determinant of HS-induced Pol III transcriptional memory

A well-documented component of the HSR is a mechanism that protects cells from repeated stress. The initial HS acts as a primer such that subsequent stresses elicit an accelerated response, enabling faster protection for the cell (Liu *et al.* 2018). This type of memory has been observed for HSF1-mediated Pol II transcription of HSP genes (Vihervaara *et al.* 2021), but how such protective mechanisms affect Pol III transcription is unknown. To better understand the regulation of tRNA transcription during HS memory, we performed an initial HS by exposing cells to 42°C for 60 min (preconditioning) followed by recovery at 37°C for 24 h. We then performed Pol III CUT&Tag analysis after a secondary HS exposure for varying lengths of time, from no secondary stress (PHS0) to 10 (PHS10), 30 (PHS30), and 60 min (PHS60) (Fig. 4a). In preconditioned cells, we observed that Pol III occupancy begins to be downregulated at HS10-treated cells and that this decrease persisted in HS30- and HS60-treated cells at individual loci and at all tRNA genes (Fig. 4b and c; Supplementary Fig. 5a). To compare the change in Pol III binding in preconditioned vs nonpreconditioned samples, we calculated the log$_2$ ratio of HS30 and HS60 over HS0 at all tRNA genes and performed the same analyses for all preconditioned samples (Fig. 4d). The recovery in Pol III binding after HS60 is evident in the significant difference between the log$_2$ ratios of HS30/HS0 and HS60/HS0, in contrast to the lack of change in log$_2$ ratios between PHS10/PHS0 and PHS60/PHS0

(Fig. 4d). These findings suggest that HS memory prolongs the decrease in Pol III transcription.

We then questioned the role of HSF1 on Pol III transcription of tRNAs during HS memory by performing the HS memory paradigm (Fig. 4a) in *Hsf1*$^{-/-}$ mESCs. We observed that Pol III occupancy increases in PHS10 compared to PHS0 *Hsf1*$^{-/-}$ cells at individual loci and at all tRNA genes (Fig. 4b and c; Supplementary Fig. 5b), in contrast to the pattern observed in the wild-type cells. This increase in Pol III occupancy also persists for PHS30- and PHS60-treated *Hsf1*$^{-/-}$ mESCs (Fig. 4b and c; Supplementary Fig. 5b). As in wild-type cells, we calculated log$_2$ ratio of HS over HS0 in the non- and preconditioned *Hsf1*$^{-/-}$ mESCs (Fig. 4d). Without HSF1, not only does Pol III regulation become decoupled with prolonged HS, as observed by the lack of recovery in HS60/HS0, but also with transcriptional memory during repeated stress, as observed by the vastly different patterns in PHS10–PHS60/PHS0 (Fig. 4d). Taken together, our results show that HSF1 plays an important role in Pol III regulation during HS and that knockout of HSF1 leads to not only a dysfunctional HSR but also the inability to modulate an appropriate response to recurring HS stress.

## Discussion

In this study, we demonstrate that Pol III transcription is dynamically regulated in response to HS stress. Specifically, we observe an initial reduction in Pol III gene expression during early HS, followed by an unexpected recovery under prolonged stress. Furthermore, we

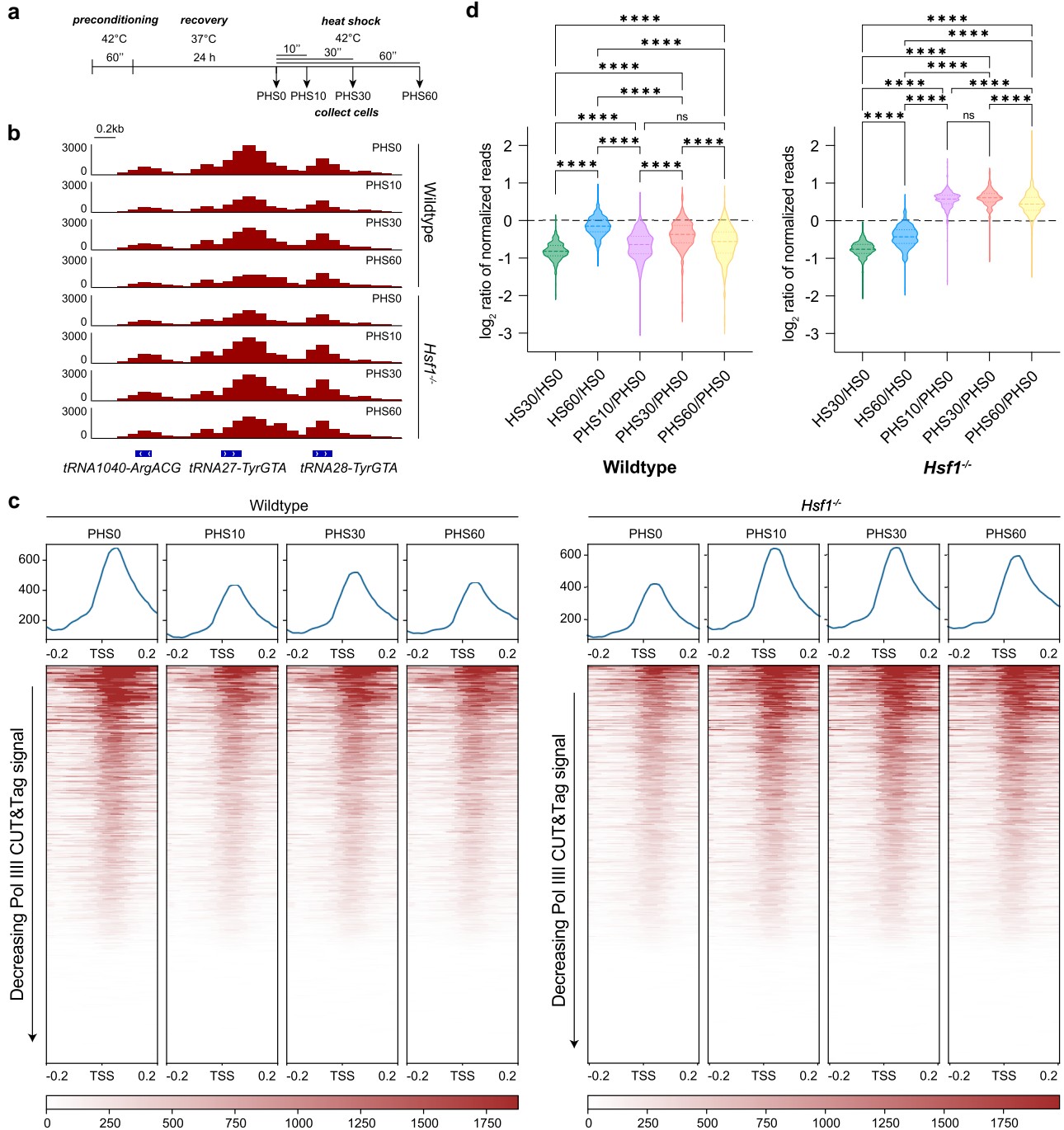

**Fig. 4.** HS preconditioning impacts tRNA regulation in wild-type and *Hsf1*⁻/⁻ mESCs. a) Experimental setup for preconditioning wild-type and *Hsf1*⁻/⁻ mESCs. Cells were treated with a single 1-h HS at 42°C, recovered for 24 h at 37°C, and collected for Pol III CUT&Tag after an additional HS treatment at 42°C for 0 (PHS0), 10 (PHS10), 30 (PHS30), or 60 min (PHS60). b) Gene browser tracks at *tRNA1040-ArgACG*, *tRNA27-TyrGTA*, and *tRNA28-TyrGTA* for reads from Pol III CUT&Tag (maroon, n = 2) in preconditioned wild-type and *Hsf1*⁻/⁻ mESCs after 0, 10, 30, or 60 min of HS treatment. c) Genome-wide average plots (top) and heatmaps (bottom) for Pol III CUT&Tag signal in a 400-bp window surrounding the TSS of all tRNA genes, arranged by decreasing signal in preconditioned wild-type (left) and *Hsf1*⁻/⁻ (right) mESCs after HS0, HS10, HS30, and HS60 treatments. d) Violin plots depicting the fold change of normalized Pol III CUT&Tag read counts from the TSS to the TES of all tRNA genes for the listed HS treatments in wild-type (left) and *Hsf1*⁻/⁻ (right) mESCs. Statistical analysis was performed using 1-way ANOVA. ns, nonsignificant. ****$P \leq 0.0001$.

show that HSF1, the master regulator of the HSR, is critical for this recovery of Pol III genes during late HS and plays a role in modulating HS-induced transcriptional memory of tRNA genes.

What drives the upregulation of tRNAs during the late stages of HS? Previous studies indicate that prolonged HS is accompanied by an increasing number of Pol II-upregulated genes (Mahat *et al.* 2016). Consequently, enhanced tRNA transcription may be

necessary to meet the increased demand for protein synthesis during sustained HS. The importance of tRNAs is underscored by their depletion during mid-HS, which coincides with active translocation and degradation of tRNAs in the nucleus (Whitney *et al.* 2007; Miyagawa *et al.* 2012; Schwenzer *et al.* 2019). Additionally, mature tRNAs are cleaved upon HS, further depleting the tRNA pool and highlighting the need for replenishing tRNAs as the HSR progresses

(Fu *et al.* 2009). The recovery of Pol III-transcribed tRNAs could represent an adaptive response, with Pol III activity rebounding from its HS-induced downregulation after prolonged stress (Abravaya *et al.* 1991). These studies suggest that the recovery of tRNA is an important step in the adaptive HSR to maintain proteostasis. Notably, this dynamic recovery contrasts with other stress responses, such as oxidative stress or nutrient deprivation, where Pol III downregulation persists over extended periods (Michels *et al.* 2010; Orioli *et al.* 2016; Torrent *et al.* 2018). Our findings suggest that this dynamic regulation and recovery of Pol III may be exclusive to certain stress responses, such as the HSR.

Although the role of HSF1 in Pol II-mediated HSR is well documented, its role in Pol III transcription during HS has been largely unexplored. Here, we show that HSF1 is not only important for the recovery of Pol III during late HS but is also essential for modulating HS-induced Pol III transcriptional memory. In *Hsf1*$^{-/-}$ mESCs, we observed a significant impairment in the recovery of Pol III binding and tRNA transcription after 60 min of HS. This result suggests that, beyond its canonical role in regulating HSP genes, HSF1 is instrumental in the adaptive recovery of Pol III-transcribed tRNA genes during HS.

HS-induced transcriptional memory is a well-documented feature of the HSR, enabling cells to respond more rapidly to subsequent HS events. This phenomenon suggests that HS can establish transcriptional memory to accelerate protective molecular mechanisms (Liu *et al.* 2018; Vihervaara *et al.* 2021). Our study is the first to document a potential HS-induced transcriptional memory for Pol III transcription. Unlike the response to an initial stress, where Pol III is first downregulated but then recovers after 60 min, subsequent stresses lead to prolonged downregulation of Pol III transcription (Fig. 4). It is possible that an eventual recovery would occur and that our experimental condition did not capture a delayed recovery. But why might such prolonged downregulation and/or delayed recovery be advantageous for subsequent stresses? One potential advantage is to increase the time for the cell to handle cumulative protein damage remaining from the initial stress. Future studies in this direction will provide further insight into the role of Pol III regulation during HS-induced memory.

The role of HSF1 in establishing HS memory for Pol III genes has been unclear. Our study shows that HSF1 plays an important role in this process. In preconditioned *Hsf1*$^{-/-}$ mESCs, tRNAs exhibited dysregulated expression during HS memory, with Pol III occupancy increasing after preconditioning rather than continuing its expected downregulation. This observation raises intriguing questions about the underlying mechanism. On one hand, HSF1 is required for Pol III recovery in nonpreconditioned *Hsf1*$^{-/-}$ mESCs. On the other hand, in preconditioned *Hsf1*$^{-/-}$ mESCs, Pol III occupancy paradoxically increases after 10 min of HS. These contrasting findings suggest that HS attenuation and transcriptional memory are distinct yet interconnected regulatory mechanisms of the HSR, both of which appear to depend on HSF1 in complex and opposing ways. These findings underscore the intricate and dynamic nature of cellular adaptation to unfavorable environments. Further investigation into Pol III-interacting factors, such as MAF1—a key repressor of Pol III transcription that functions through the mTOR pathway—may provide additional insight into the regulatory mechanisms underlying this phenomenon.

# Materials and methods
## Cell lines
For all cell lines used in this study, the parental line is JM8.N4 mouse ES cells, purchased from KOMP repository, RRID: CVCL_J962. For all experiments, a mouse *Hsf1*$^{-/-}$ ES cell line (Price *et al.* 2023) or the mouse ES cell line C64 was used. The *Hsf1*$^{-/-}$ mES cell line is a genetically modified JM8.N4 cell line containing a full deletion of the *Hsf1* gene locus as previously described (Price *et al.* 2023). C64 is a CRISPR-Cas9 genetically modified JM8.N4 cell line containing mAID-TBP knockin obtained as previously described (Kwan *et al.* 2023).

## Cell culture
Mouse ES cells were cultured on 0.1% gelatin-coated plates in mESC media: KnockOut DMEM (Corning) with 15% FBS (Cytiva HyClone), 0.1 mM MEM nonessential amino acids (Gibco), 2 mM GlutaMAX (Gibco), 0.1 mM 2-mercaptoethanol (Sigma-Aldrich), 100-U/mL penicillin (Cytiva HyClone), 100μg/mL streptomycin (Cytiva HyClone), and 1,000 units/mL of ESGRO Recombinant Mouse LIF Protein (Chemicon). Mouse ES cells were fed daily, cultured at 37°C in a 5% $CO_2$ incubator, and passaged every 2 days by trypsinization. For MG132 treatment, mESCs were treated with DMSO or 10 uM of MG132 for 0, 120, and 240 min. HS was performed at 42°C in a 5% $CO_2$ incubator for either 30 or 60 min. For preconditioned cells, mESCs were heat shocked for 1 h at 42°C, recovered at 37°C for 24 h, and subjected to a single round of HS for 10, 30, or 60 min.

## Antibodies for immunoblot
Primary antibodies are as follows: α-Tubulin 1:7,000 (Abcam Cat# ab6046, RRID: AB_2210370), α-RPC7 1:2,000 (Santa Cruz Biotechnology Cat# sc-21754, RRID:AB_675824), and α-HSF1 1:1,000 (Abcam Cat# ab2923, RRID:AB_303419). Secondary antibodies are as follows (1:15,000): IRDye 800CW Goat anti-Mouse IgG (LI-COR Biosciences Cat# 926-32210, RRID: AB_621842) and IRDye 680RD Goat anti-Rabbit IgG (LI-COR Biosciences, Cat# 926-68071, RRID: AB_10956166).

## CUT&Tag
CUT&Tag was performed as previously described (Kaya-Okur *et al.* 2019), but with the following modifications. Cells were harvested at room temperature and 100,000 mESCs were used per sample. Antibodies used include α-RPC7 (Pol III) (Santa Cruz Biotechnology Cat# sc-21754, RRID: AB_675824), Rabbit anti-Mouse IgG (Abcam Cat# ab46540, RRID: AB_2614925), and α-H3K27me3 as library positive control (Cell Signaling Technology Cat# 9733, RRID: AB_2616029). Secondary antibodies used include Guinea Pig α-Rabbit IgG (Antibodies-Online Cat# ABIN101961, RRID: AB_10775589) and Rabbit α-Mouse IgG (Abcam Cat# ab46540, RRID: AB_2614925). Secondary antibody incubation times were 1 h at room temperature. pA-Tn5 was produced in-house by the UBC Biomedical Research Centre, and the pA-Tn5 adapter complex was added at a final concentration of 1:20. Sequencing was performed at the UBC Biomedical Research Centre using NextSeq2000 with 50 cycles.

## CUT&Tag analysis
Reads were mapped to the mm10 genome using Bowtie2 v2.4.2 (Langmead and Salzberg 2012) with the following parameters: --no-unal --local --very-sensitive-local --no-discordant –no-mixed --phred33 –I 10 –X 2000. The resulting SAM files were converted to BAM files using SAMtools v1.7 (Li *et al.* 2009). To normalize the Pol III CUT&Tag data, we used ChIP-seq-Spike-In-Free, a normalization method to determine scaling factors for samples across various conditions (Jin *et al.* 2020). BigWig coverage files were generated from the BAM files using deepTools v3.5.0 (Ramírez *et al.* 2016) and the normalization factors calculated

by ChIP-seq-Spike-In-Free. Biological replicates were merged using bigWigMerge (Kent *et al.* 2010), which sums the reads after normalizing each library using ChIP-seq-Spike-In-Free. Heatmaps and average plots were generated with deepTools v3.5.0, and gene plots were mapped using the Integrative Genomics Viewer (IGV) (Thorvaldsdottir *et al.* 2013). Read counts were generated with bedTools (Quinlan and Hall 2010), normalized with the scaling factors calculated by ChIP-seq-Spike-In-Free, and analyzed with GraphPad Prism.

## NET-seq analysis

Reanalysis of NET-seq data (GSE172401) was performed as previously described (Kwan *et al.* 2023). Read counts across tRNAs were generated from *Drosophila* spike-in normalized bam files using bedTools and analyzed with GraphPad Prism.

## PRO-seq analysis

PRO-seq data from Vihervaara *et al.* 2021 (GSE128160) were reanalyzed for wild-type MEFs (unstressed, HS25, HS60). PRO-seq fastq files were preprocessed and mapped to the mm10 genome using the proseq2.0 pipeline (Chu *et al.* 2019). BigWig coverage files were generated with BAM files using deepTools v3.5.0 and normalized by using PRO-seq reads from the 3′ end of long genes > 400 kb as previously described (Mahat *et al.* 2016). Normalized replicates were merged using bigWigMerge (Kent *et al.* 2010). Heatmaps and average plots were generated with deepTools v3.5.0, and gene plots were mapped using IGV.

## BRI-qPCR

BRI-qPCR was adapted from the BRIC-seq protocol published previously (Imamachi *et al.* 2014). The same number of cells for each sample was plated and cultured until ~90% confluency. All cells were BrU labeled at a final concentration of 1 mM for a total of 30 min. For HS30 treatment, BrU was added at a final concentration of 1 mM, and cells were promptly heat shocked at 42°C in a 5% $CO_2$ incubator for 30 min. For HS60 treatment, cells were heat shocked at 42°C in a 5% $CO_2$ incubator for 30 min, labeled with BrU (final concentration: 1 mM), and continued to be incubated at 42°C in a 5% $CO_2$ incubator for 30 more minutes, totaling 60 min of HS. To normalize signal across all samples, 2.5 µg of in vitro transcribed 5-BrU-labeled RNA encoding for NanoLuc Luciferase was spiked into each well, followed by TRIzol extraction. BrU-labeled samples were incubated at 80°C for 2 min, followed by immediate cooling in an iced-water bath. Protein G Dynabeads conjugated with α-BrU antibody (MBL International Cat# MI-11-3, RRID: AB_590678) were prepared as published previously (Imamachi *et al.* 2014). BrU-labeled samples were then incubated with the conjugated beads at 4°C for 2 h and washed 6 times with ice-cold BSA/Triton/PBS. BrU-labeled RNAs were eluted in 90-µL 10 mM Tris–HCl at pH 7.4 and 6.25 mM EDTA and isolated via TRIzol LS Reagent according to the manufacturer's protocol and resuspended in 25 µL of DEPC-treated water. The entire sample was DNase treated using the Promega RQ1 RNase-Free DNase Kit (M6101) and reverse transcribed using the New England BioLabs LunaScript RT SuperMix Kit (E3010), according to the manufacturer's protocol. The resulting cDNAs were diluted 1:5, and 4.5 µL of sample was used for each qPCR reaction with Luna Universal qPCR Master Mix (M3003) with the QuantStudi 3 Real-Time PCR System.

## RT-qPCR

Cells were cultured until ~90% confluency on tissue culture-treated plates at 37°C in a 5% $CO_2$ incubator. HS was performed at 42°C in a 5% $CO_2$ incubator for either 30 or 60 min. For MG132 treatment, mESCs were treated with DMSO or 10 uM of MG132 for 0, 120, and 240 min. After the indicated treatments, cells were washed with 1× PBS, trypsinized, and pelleted by centrifugation at 600 *g*. 2.5 µg of in vitro transcribed 5-BrU-labeled RNA encoding for NanoLuc Luciferase was spiked in to each sample to normalize signal across all samples. RNA was extracted from the pellet by TRIzol extraction, and RNA concentrations were measured by Nanodrop. One microgram of RNA was then DNAse treated using the Promega DNAse Kit (M6101), and the resulting DNAse-treated RNA was reverse transcribed using the New England BioLabs LunaScript RT SuperMix kit (E3010). cDNA was diluted to 1 ng/µL and 4.5 ng of cDNA was used in each reaction using the New England BioLabs Luna Universal qPCR Master Mix Kit (M3003) with the QuantStudio 3 Real-Time PCR System.

## Data availability

The data that support the findings of this study are openly available in Gene Expression Omnibus reference number GSE282678.
Supplemental material available at GENETICS online.

## Acknowledgments

This work was supported by Life Sciences Institute Cores (LSI Imaging, ubcFLOW, and qPCR Core) and by the UBC GREx Biological Resilience Initiative. We thank T. Stach (School of Biomedical Engineering Sequencing Core, UBC) for Illumina sequencing and Dr. Eric Jan for insightful comments on the manuscript. We thank Heather Baker and Dr. Thibault Mayor for generously providing MG132. TFN, JZJK, and JHC are supported by the UBC 4-Year Fellowship. SST is a Canada Research Chair Tier 2 in Mechanisms of Gene Regulation and is supported by the Michael Smith Foundation for Health Research.

## Funding

This work was supported by the Canadian Institutes of Health Research Project Grant award to SST (PJT-162289), the Natural Sciences and Engineering Research Council of Canada Grant award to SST (RGPIN-2020-06106), and the Stem Cell Network to SST (AWD-021244).

## Conflicts of interest

The authors declare no conflicts of interest.

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

*Editor: O. Rando*