## [Peer Review File · Genetics]

Dynamic regulation of murine RNA Polymerase III transcription during heat shock stress

Thomas Nguyen, James Kwan, Jennifer Mitchell, Jieying Cui, and Sheila Teves

NOTE: The reviews and decision letters are unedited and appear as submitted by the reviewers.

In extremely rare instances and as determined by a Senior Editor or the EIC, portions of a review may be redacted. If a review is signed, the reviewer has agreed to no longer remain anonymous.

The review history appears in chronological order.

Review Timeline:

Submission Date:	2024-11-30
Editorial Decision:	2025-01-13
Resubmission Received:	2025-02-18
Accepted:	2025-02-27

January 13, 2025

GENETICS-2024-307669

Dynamic regulation of murine RNA Polymerase III transcription during heat shock stress

Dear Dr. Teves:

Three experts in the field have reviewed your manuscript, and I am pleased to inform you that, with minor revisions, it is potentially suitable for publication in GENETICS. The reviewers have comments and concerns that need to be addressed in a revised manuscript. You can read their reviews at the end of this email.

Reviewer 1 has the most extensive requests for additional work, some of which in my opinion go beyond absolutely necessary experiments. That said, point 2 only requires additional data analysis, and I think it is reasonable to ask whether MG132 induces a stress response here. As for showing whether this occurs in another cell type (Point 3), this would be a great addition if you happen to have the experiment done, or were planning on it -- it would be valuable to include -- but I do not think this is essential. Please also address the minor comments from Reviewers 2 and 3, and Reviewer 1 point 1.

We look forward to receiving your revised manuscript. Please let the editorial office know approximately how long you expect to need for revisions.

Upon resubmission, please include:

1. A clean version of your manuscript;
2. A marked version of your manuscript in which you highlight significant revisions carried out in response to the major points raised by the editor/reviewers (track changes is acceptable if preferred);
3. A detailed response to the editor's/reviewers' comments and to the concerns listed above. Please reference line numbers in this response to aid the editors.

Additionally, please ensure that your resubmission is formatted for GENETICS.

<https://academic.oup.com/genetics/pages/general-instructions>

Follow this link to submit the revised manuscript: Link Not Available

Sincerely,

Oliver Rando
Associate Editor
GENETICS

Approved by:
Karen Arndt
Senior Editor
GENETICS

Reviewer #1 :

Nguyen et al. present an interesting study examining how Pol III occupancy and tRNA transcription are regulated in mESCs during heat shock (HS). Most previous work has focused on Pol II transcription under stress conditions, this manuscript addresses a relatively understudied aspect, the regulation of Pol III-driven tRNA transcription under HS. By employing CUT&Tag approaches and re-analyzing previously published PRO-seq data, the authors reveal that Pol III occupancy and tRNA transcription decline after a 30 minutes of HS but recover after a prolonged (60-minute) HS. Moreover, this recovery is dependent on HS transcription factor, HSF1. Additionally, the study investigates preconditioning affects the response. Following a second round of HS, Pol III-driven tRNA transcription decreases in WT cells, but surprisingly increases in hsf1^{-/-} cells. These findings suggest that HSF1 is not only important for initial adaptation but also for shaping transcriptional memory upon recurrent stress. The authors present detailed and genome wide evidence and support their conclusions at individual tRNA loci and a metagene level. Nevertheless, certain aspects of the paper are insufficiently explored. For instance, the study provides limited mechanistic insight into how Pol III activity is modulated, such as how HSF1 modulates Pol III binding to tRNA genes. Additionally, the

authors focus exclusively on mESCs. Expanding the analysis to other cell types could increase the generality and impact of these findings. Addressing these issues would make the manuscript more suitable for publication.

Major issues:

1. In the preconditioning experiments, the authors show that WT cells exhibit a marked reduction in Pol III occupancy at 10, 30, and 60 minutes after the second HS exposure, without the rebound observed during the initial stress response. Given that HSF1 is proposed to be critical for recovery, the authors should discuss why the tRNA transcription does not recover at 60 minutes in the preconditioned scenario and how HSF1's role might differ between initial and subsequent rounds of HS.
2. While the authors establish that Pol III occupancy changes during HS and depends on HSF1, the underlying mechanisms remain unknown. Does Pol III shift to different genomic regions other than tRNA and ncRNAs upon HS? Is the pol III transcriptional machinery functionally altered by HSF1? Are there direct interactions between HSF1 and Pol III? Additional experiments or data analysis will help the mechanistic investigation.
3. While mESCs is a good model, it would greatly enhance the generality of the paper if the authors could show that this regulatory pattern of pol III and tRNA also occurs in other cell types.
4. Many proteasome inhibitors could also activate HSF1. Testing whether similar patterns of Pol III-driven tRNA transcription occur when cells are treated with these proteotoxic agents, such as MG132, would help determine if the observed phenomenon are unique to HS or represent a more general cellular stress response.

Minor issues:

1. It is nice to clarify the rationale behind selecting the 30- and 60-minute time points for HS in the manuscript.
2. Critical results in the paper, pol III occupancy and tRNA transcription upon HS in WT and Hsf1^{-/-} cells are shown in the separate figures (figure 1 and figure 2). Presenting side-by-side comparisons would allow readers to visualize the data more easily.

Reviewer #2 :

This manuscript by Thomas Nguyen and colleagues in the Teves lab investigates RNA polymerase III transcription during heat stress. Under heat stress, cells undergo a global shift in gene expression, suppressing most transcription and translation while favoring the production of chaperone proteins to mitigate protein damage. This study investigates how RNA Polymerase III (Pol III), the enzyme responsible for transcribing tRNAs and other small RNAs, is regulated during heat stress in mouse embryonic stem cells (mESCs). The authors report a biphasic response in Pol III transcription: an initial downregulation after 30 minutes of heat shock, followed by an upregulation at 60 minutes. Interestingly, the early suppression of Pol III activity occurs independently of HSF1, the primary regulator of the heat shock response, whereas the later increase in Pol III transcription depends on HSF1. These findings reveal a dynamic and adaptive Pol III response to heat stress and underscore the nuanced interplay between tRNA expression and the canonical heat shock response.

Overall, this study is an excellent contribution to the field. The manuscript is concise, clearly written, and effectively communicates the key findings. Analyses of Pol III CUT&Tag data alongside existing NET-seq and PRO-seq datasets provide complementary approaches that nevertheless provide consistent conclusions, strengthening the study's claims. All analyses are well described and appear to be executed properly. The use of HSF1-deficient (HSF1^{-/-}) cells to dissect the role of HSF1 in heat-induced Pol III transcriptional changes is a lovely and compelling addition to the manuscript. This experiment elegantly demonstrates that while early Pol III suppression is HSF1-independent, the later increase in transcription critically depends on HSF1, providing important mechanistic insights into the regulation of Pol III during heat stress.

If I had one comment, it's that I found the striking differences in Pol III transcription between HSF1^{-/-} and wild type in heat stress preconditioned cells both surprising and interesting. It would be amazing to dig into this more. I also understand, of course, that this is almost certainly beyond the scope of the present work, and the current manuscript stands on its own.

Reviewer #3 :

The manuscript "Dynamic regulation of murine RNA Polymerase III transcription during heat shock stress" describes the regulation of RNA polymerase III transcription after heat shock of mouse cells. While the effects of heat shock on RNA polymerase II-dependent transcription have been extensively characterized, this manuscript shows that effects on RNAPIII are different. RNAPIII binding at tRNA genes is moderately reduced immediately after heat shock begins, but recovers by 60 minutes. While it can be difficult to normalize experiments with global effects, the authors use NET-seq, Pro-seq, and metabolic labeling to confirm these effects. The authors go on to show that the transcriptional recovery from extended heat shock and the memory of past heat shocks depends on the Hsf1 transcription factor, based on analysis of Hsf1 mutant cells. On this last point, I was unclear on what exactly the authors are inferring - do the authors think these are direct of Hsf1 binding at these genes, or indirect effects of Hsf1 at other genes that mediate these effects? The first paragraph on page 8 discusses and points out the

similarity of effects for RNAPII- and for RNAPIII-dependent genes. But for many of these RNAPII-dependent genes, HSF may bind at their promoters. Could the authors comment on if Hsf1 is present at RNAPIII-dependent genes?

Associate Editor Comments:

We thank the reviewers for their thorough reading of our manuscript and their insightful comments. Below, please find our point-by-point response to the individual reviewer's comments.

Reviewer #1 :

Nguyen et al. present an interesting study examining how Pol III occupancy and tRNA transcription are regulated in mESCs during heat shock (HS). Most previous work has focused on Pol II transcription under stress conditions, this manuscript addresses a relatively understudied aspect, the regulation of Pol III-driven tRNA transcription under HS. By employing CUT&Tag approaches and re-analyzing previously published PRO-seq data, the authors reveal that Pol III occupancy and tRNA transcription decline after a 30 minutes of HS but recover after a prolonged (60-minute) HS. Moreover, this recovery is dependent on HS transcription factor, HSF1. Additionally, the study investigates preconditioning affects the response. Following a second round of HS, Pol III-driven tRNA transcription decreases in WT cells, but surprisingly increases in *hsf1*^{-/-} cells. These findings suggest that HSF1 is not only important for initial adaptation but also for shaping transcriptional memory upon recurrent stress. The authors present detailed and genome wide evidence and support their conclusions at individual tRNA loci and a metagene level. Nevertheless, certain aspects of the paper are insufficiently explored. For instance, the study provides limited mechanistic insight into how Pol III activity is modulated, such as how HSF1 modulates Pol III binding to tRNA genes. Additionally, the authors focus exclusively on mESCs. Expanding the analysis to other cell types could increase the generality and impact of these findings. Addressing these issues would make the manuscript more suitable for publication.

We thank the reviewer for their thoughtful and constructive assessment of our work. We appreciate their recognition of the novelty and significance of our findings, as well as their enthusiasm for further mechanistic exploration and broader applicability across cell types.

While we acknowledge the importance of these directions, we believe that a detailed mechanistic dissection of HSF1's regulation of Pol III activity and an extension to other cell types are beyond the scope of this study, particularly given the short report format. Addressing these aspects would require substantial additional experimentation that, while valuable, is not feasible within the constraints of this manuscript. Given that our lead author has recently graduated and is moving on, pursuing these avenues would be more appropriate for a future, expanded study.

Nevertheless, we believe that our findings provide new and meaningful insights into the regulation of Pol III-driven transcription under stress, which merit publication in the current short report form. Below, we provide a point-by-point response to the specific comments raised.

Major issues:

1. In the preconditioning experiments, the authors show that WT cells exhibit a marked reduction in Pol III occupancy at 10, 30, and 60 minutes after the second HS exposure, without the rebound observed during the initial stress response. Given that HSF1 is proposed to be critical for recovery, the authors should discuss why the tRNA transcription does not recover at 60 minutes in the preconditioned scenario and how HSF1's role might differ between initial and subsequent rounds of HS.

We agree and have included the following in the Discussion section: “Our study is the first to document a potential HS-induced transcriptional memory for Pol III transcription. Unlike the response to an initial stress, where Pol III is first down-regulated but then recovers after 60 minutes, subsequent stresses lead to prolonged downregulation of Pol III transcription (Figure 4). It is possible that an eventual recovery would occur, and that our experimental condition did not capture a delayed recovery. But why might such prolonged downregulation and/or delayed recovery be advantageous for subsequent stresses? One potential advantage is to increase the time for the cell to handle cumulative protein damage remaining from the initial stress. Future studies in this direction will provide further insight into the role of Pol III regulation during HS-induced memory.” (lines 359-367)

2. While the authors establish that Pol III occupancy changes during HS and depends on HSF1, the underlying mechanisms remain unknown. Does Pol III shift to different genomic regions other than tRNA and ncRNAs upon HS? Is the pol III transcriptional machinery functionally altered by HSF1? Are there direct interactions between HSF1 and Pol III? Additional experiments or data analysis will help the mechanistic investigation.

As mentioned above, we hope to pursue the detailed mechanisms of HSF1's role in Pol III transcription in future studies. Indeed, we have begun to test for indirect effects of HSR into Pol III regulation (see our response to Reviewer 2's comment below). These are still exploratory, and would require substantial time and resources to pursue in future studies.

That said, further analysis of existing data sets, including those from previously published studies and ones generated here, have provided additional insights. First, we looked for potential global differences between Pol III CUT&Tag in different conditions by performing Pearson correlation analysis of the signal throughout the genome in 10kb bins. The strong correlation among Pol III samples during heat shock, regardless of HSF1 presence, suggests that the Pol III signal throughout the genome is not distinctly different among the samples. This implies that Pol III is not redistributed away from gene targets and to other genomic regions, but rather only the signal strength is changing.

Peak calling with CUT&Tag data is not quite as established as ChIP-seq pipelines. Using traditional peak callers such as MACS with CUT&Tag data generates large numbers of false positive peaks, as the comparative IgG data sets are too sparse to be used as background for these peak callers. Therefore, we cannot be confident with called peaks that would be outside of Pol III target genes.

As for the relationship between HSF1 and Pol III machinery, the reviewer brings up an interesting question that we hope to pursue in the future. Since the initial downregulation of Pol III upon heat shock is HSF1-independent, we predict that HSF1 does not directly interact with the Pol III machinery. Indeed, previously published HSF1 IP-MS data does not identify Pol III machinery proteins as interacting partners (Smith et al. 2022; PMID: 35294249). Furthermore, evidence for HSF1 binding at tRNA genes is quite weak. Using previously published datasets of HSF1 ChIP-seq in MEFs (Himanen et al. 2022; PMID: 35687139, panel A) and HSF1 CUT&Tag in mESCs (Price et al. 2023; PMID: 37114996, panel B), we plotted the average signal and

heatmaps for HSF1 on all tRNA genes, along with the respective IgG negative controls. Although the HSF1 signal at tRNA genes is somewhat above the IgG background, this signal is orders of magnitude lower than the signal observed for Pol III at tRNA genes. This is evident in the IgV tracks (panel C) showing the difference in dynamic range among the samples analyzed, where Pol III ranges from 0 to 2290, whereas the HSF1 CUT&Tag in mESCs ranges from 0 to 72. These data suggest that HSF1 does not directly regulate Pol III.

Although we do not have evidence supporting a direct HSF1-Pol III interaction, we show that the recovery of Pol III transcription during prolonged HS is HSF1-dependent. We hypothesize that this connection is indirect. That is, as cells experience prolonged heat shock, HSF1 activates gene target(s) that then feedback to Pol III transcription. To fully test this hypothesis, we would need to perform a systematic screen for late targets of HSF1 rather than a candidate-based approach (see our response to Reviewer 3's comment). This undertaking would be quite beyond the scope of this current manuscript but one we are excited to pursue in the future.

3. While mESCs is a good model, it would greatly enhance the generality of the paper if the authors could show that this regulatory pattern of pol III and tRNA also occurs in other cell types.

We appreciate the reviewer's suggestion to examine whether the observed regulation of Pol III and tRNA transcription extends to other cell types. While our study primarily focuses on mESCs, we have aimed to assess the generality of our findings by analyzing previously published PRO-seq datasets from other models.

As presented in Figure 1, we reanalyzed PRO-seq data from mouse embryonic fibroblasts (MEFs) published by Vihervaara et al. (2021; PMID: 33784494). Although we did not generate these data ourselves, the experimental conditions in the Vihervaara study closely align with our own, allowing for meaningful comparisons. The consistency of the results between mESCs and MEFs is encouraging, particularly given that the original study did not specifically focus on tRNA transcription.

To further explore this phenomenon in human cells, we analyzed PRO-seq datasets from studies that examined the heat shock response in immortalized cancer cell lines. While these datasets provide additional insights, they also present inherent limitations. Specifically, these studies only include data from a 60-minute heat shock time point, preventing us from fully assessing the transcriptional dynamics observed in mESCs and MEFs. The figure below summarizes data from three studies:

1. **Cardiello et al. 2024 (PMID: 39116081)**: PRO-seq analysis in lymphoblastoid cells revealed an increase in tRNA expression following 60 minutes of heat shock, consistent across two biological replicates.
2. **Dastidar et al. 2023 (PMID: 37973875)**: PRO-seq analysis in MCF7 cells showed inconsistent results between replicates—one indicated a decrease in tRNA expression after 60 minutes of heat shock, while the other showed a slight increase.
3. **Vihervaara et al. 2021 (PMID: 33784494)**: PRO-seq analysis in K562 cells at 0, 30, and 60 minutes of heat shock revealed an increase in tRNA expression at both 30 and 60 minutes. However, this dataset did not capture the transient decline in tRNA expression at 30 minutes that we observed in mESCs.

Wildtype human lymphoblastoid cells (Cardiello et al., 2024, PMID: 39116081)

Wildtype MCF-7 cells (Dastidar et al., 2023, PMID: 37973875)

Wildtype K562 cells (Vihervaara et al., 2021, PMID: 33784494)

These datasets highlight some of the challenges in comparing heat shock responses across different cell types and species. Notably, cancer cells exhibit fundamental differences in stress response pathways, including elevated HSF1 activity (Mendilo et al. 2012; PMID: 22863008), which could influence Pol III regulation and transcriptional dynamics. Given these confounding factors, we opted not to include these analyses in the main manuscript but provide them here in response to the reviewer's comment.

In addition to these cancer cell datasets, we analyzed TT-seq data from a study by Cugusi et al. (2022; PMID: 35114099), which examined heat shock responses in non-cancerous human fibroblasts at 0, 30, 60, and 180 minutes. Our analysis of tRNA transcription in this dataset reveals an initial induction at 30 minutes, a slight decline at 60 minutes, and a pronounced increase at 180 minutes, shown below. While these dynamics do not align perfectly with those observed in mESCs, they suggest that similar regulatory mechanisms may be at play in human fibroblasts.

Overall, while these published datasets provide some evidence that the regulatory patterns we observed in mESCs extend to other cell types, differences in experimental conditions, cell type-specific stress responses, and cancer-related alterations complicate direct comparisons. We appreciate the reviewer's suggestion and have incorporated these additional analyses here to provide a broader context for our findings.

4. Many proteasome inhibitors could also activate HSF1. Testing whether similar patterns of Pol III-driven tRNA transcription occur when cells are treated with these proteotoxic agents, such as MG132, would help determine if the observed phenomenon are unique to HS or represent a more general cellular stress response.

We appreciate the reviewer's insightful comment regarding the activation of HSF1 by proteasome inhibitors. To determine whether the observed changes in Pol III-driven tRNA transcription are specific to heat shock or part of a broader cellular stress response, we treated mESCs with the proteasome inhibitor MG132 for 2 and 4 hours (two biological replicates) and assessed transcriptional changes by RT-qPCR (data now presented in Figure S2F).

As expected, we observed a strong induction of Hsp70 following 2 hours of MG132 treatment, which further increased at 4 hours. This result is consistent with previous findings demonstrating that MG132 activates HSF1 (Kim et al. 2011; PMID: 21738571). However, analysis of AlaAGC and TyrGTA tRNA expression revealed no significant changes over the course of MG132 treatment. Notably, the time points selected for MG132 treatment align with those commonly used in the literature.

These findings indicate that while proteasome inhibition via MG132 robustly activates HSF1 and induces heat shock protein expression, it does not significantly affect Pol III regulation of tRNA transcription. We have incorporated these findings into the manuscript (Figure S2F) and added the following paragraph to the text:

“To determine whether the observed changes in tRNA transcription are specific to heat stress, we treated mESCs with the proteasome inhibitor MG132 for 0, 120, and 240 minutes. MG132 has been shown to activate HSF1 and induce heat shock protein expression in mammalian cells (Kim et al. 2011). We confirmed this effect by RT-qPCR analysis of *Hspa1a*, which showed increased expression at 120 and 240 minutes post-treatment. However, RT-qPCR analysis of *AlaAGC* and *TyrGTA* tRNAs revealed no significant changes in tRNA transcription following MG132 treatment (Figure S2F). These results suggest that the observed changes in tRNA transcription are linked to the HSR beyond the direct activation of HSF1.” (lines 229-236)

To further investigate whether the regulation of Pol III activity observed in heat shock is unique or shared among other cellular stress responses, we analyzed publicly available PRO-seq data from Himanen et al. (2022; PMID: 35687139), in which MEF cells were subjected to oxidative stress. While oxidative stress primarily activates hypoxia-induced factors, it also crosstalks with the HSR to induce HSF1 activation (Himanen et al. 2022; PMID: 35687139). This study demonstrated that although oxidative stress activates HSF1, the transcriptional programs regulated by HSF1 under oxidative stress are distinct from those induced by heat shock.

The authors performed PRO-seq on wild-type and HSF1 knockout MEFs following 2 hours of oxidative stress. We analyzed the average PRO-seq signal at all tRNA genes under control conditions (HS0), after 60 minutes of heat shock (HS60), and following 2 hours of oxidative stress (OS) in two biological replicates (rep1, rep2).

Wildtype MEFs (Himanen et al., 2022, PMID: 35687139)

Hsf1 KO MEFs (Himanen et al., 2022, PMID: 35687139)

Our analysis indicates that oxidative stress leads to a reduction in tRNA expression in both WT and HSF1 knockout cells compared to control conditions, suggesting that downregulation of Pol III activity may be a general cellular stress response that is not triggered by proteasome inhibition. However, since this study examined only a single time point (2 hours), we cannot assess the temporal dynamics of tRNA regulation under oxidative stress or directly compare them to those observed during heat shock. Therefore, we have included this analysis here as an additional point of comparison.

Minor issues:

1. It is nice to clarify the rationale behind selecting the 30- and 60-minute time points for HS in the manuscript.

We now include the following statement in the manuscript: "These time points are consistent with previous studies delineating early/intermediate from late transcriptional response related to heat shock (Mahat et al. 2016; Vihervaara et al. 2021)." (lines 179-181)

2. Critical results in the paper, pol III occupancy and tRNA transcription upon HS in WT and Hsf1^{-/-} cells are shown in the separate figures (figure 1 and figure 2). Presenting side-by-side comparisons would allow readers to visualize the data more easily.

We appreciate the reviewer's suggestion to present side-by-side comparisons of Pol III occupancy and tRNA transcription in WT and Hsf1^{-/-} cells. In our initial figure design, we attempted to combine these data into a single figure; however, this resulted in an overly complex and crowded layout that compromised readability and figure quality. To ensure clarity while maintaining the integrity of the data presentation, we opted to separate the results into two

figures. While this format does not allow for easy side-by-side comparison, we believe it provides a clearer and more accessible visualization of the findings.

Reviewer #2 :

This manuscript by Thomas Nguyen and colleagues in the Teves lab investigates RNA polymerase III transcription during heat stress. Under heat stress, cells undergo a global shift in gene expression, suppressing most transcription and translation while favoring the production of chaperone proteins to mitigate protein damage. This study investigates how RNA Polymerase III (Pol III), the enzyme responsible for transcribing tRNAs and other small RNAs, is regulated during heat stress in mouse embryonic stem cells (mESCs). The authors report a biphasic response in Pol III transcription: an initial downregulation after 30 minutes of heat shock, followed by an upregulation at 60 minutes. Interestingly, the early suppression of Pol III activity occurs independently of HSF1, the primary regulator of the heat shock response, whereas the later increase in Pol III transcription depends on HSF1. These findings reveal a dynamic and adaptive Pol III response to heat stress and underscore the nuanced interplay between tRNA expression and the canonical heat shock response.

Overall, this study is an excellent contribution to the field. The manuscript is concise, clearly written, and effectively communicates the key findings. Analyses of Pol III CUT&Tag data alongside existing NET-seq and PRO-seq datasets provide complementary approaches that nevertheless provide consistent conclusions, strengthening the study's claims. All analyses are well described and appear to be executed properly. The use of HSF1-deficient (HSF1^{-/-}) cells to dissect the role of HSF1 in heat-induced Pol III transcriptional changes is a lovely and compelling addition to the manuscript. This experiment elegantly demonstrates that while early Pol III suppression is HSF1-independent, the later increase in transcription critically depends on HSF1, providing important mechanistic insights into the regulation of Pol III during heat stress.

If I had one comment, it's that I found the striking differences in Pol III transcription between HSF1^{-/-} and wild type in heat stress preconditioned cells both surprising and interesting. It would be amazing to dig into this more. I also understand, of course, that this is almost certainly beyond the scope of the present work, and the current manuscript stands on its own.

We sincerely thank Reviewer 2 for their thoughtful assessment of our manuscript and for their positive feedback on the clarity and impact of our study. We are also intrigued by the striking differences in Pol III transcription observed in HSF1^{-/-} preconditioned cells and appreciate the reviewer's interest in further exploring this phenomenon.

In response to this and Reviewer 1's related comment, we have expanded our Discussion section to speculate on potential mechanisms underlying this effect. One possibility is that HSF1 may activate specific genes that subsequently influence Pol III regulation by 60 minutes of heat shock. A recent study by Leone et al. (2024; PMID: 38266641) demonstrated that Hsp70, a

major chaperone induced by HSF1 during heat shock, binds and regulates tRNAs and 5S rRNA. To explore whether Hsp70 might interact with Pol III in a heat shock-dependent manner, we performed co-immunoprecipitation (co-IP) experiments at 0, 30, and 60 minutes of heat shock (see figure below). However, we did not detect a direct interaction between these two proteins. While this candidate-based approach provided initial insights, a more comprehensive, large-scale investigation will be necessary to fully dissect this regulatory relationship.

Additionally, we plan to investigate MAF1, a well-established negative regulator of Pol III, to determine whether it interacts with HSF1 in the context of heat shock. Given the potential complexity of this regulatory network, we anticipate that this line of inquiry will form the basis of a future study. To reflect this, we have incorporated the following statement into the Discussion section:

"Further investigation into Pol III-interacting partners, such as MAF1—a master repressor of Pol III transcription that operates via the mTOR pathway—may provide additional insights into the mechanisms underlying this phenomenon." (lines 378-380)

Reviewer #3 :

The manuscript "Dynamic regulation of murine RNA Polymerase III transcription during heat shock stress" describes the regulation of RNA polymerase III transcription after heat shock of mouse cells. While the effects of heat shock on RNA polymerase II-dependent transcription have been extensively characterized, this manuscript shows that effects on RNAPIII are different. RNAPIII binding at tRNA genes is moderately reduced immediately after heat shock begins, but recovers by 60 minutes. While it can be difficult to normalize experiments with global effects, the authors use NET-seq, Pro-seq, and metabolic labeling to confirm these effects. The

authors go on to show that the transcriptional recovery from extended heat shock and the memory of past heat shocks depends on the Hsf1 transcription factor, based on analysis of Hsf1 mutant cells.

On this last point, I was unclear on what exactly the authors are inferring - do the authors think these are direct of Hsf1 binding at these genes, or indirect effects of Hsf1 at other genes that mediate these effects? The first paragraph on page 8 discusses and points out the similarity of effects for RNAPII- and for RNAPIII-dependent genes. But for many of these RNAPII-dependent genes, HSF may bind at their promoters. Could the authors comment on if Hsf1 is present at RNAPIII-dependent genes?

We appreciate Reviewer 3's insightful comment. Reviewer 1 brought up a similar concern (see Rev 1 comment #2), and to avoid repetition, we direct the reviewer to our response above.

In sum, we find no evidence for a direct interaction between HSF1 and Pol III machinery. Pearson correlation analysis of Pol III CUT&Tag data suggests that Pol III is not redistributed across the genome but rather changes in signal strength. While HSF1 does not appear to directly regulate Pol III, the recovery of Pol III transcription during prolonged heat shock is HSF1-dependent. We hypothesize that this is likely an indirect mechanism where HSF1 activates gene targets that subsequently influence Pol III activity. A comprehensive, systematic screen to fully test this hypothesis is beyond the scope of the present study but represents an exciting avenue for future research.

February 27, 2025

RE: GENETICS-2025-307891

Dr. Sheila S Teves
The University of British Columbia
Biochemistry and Molecular Biology
2350 Health Sciences Mall Room 5520
VANCOUVER, N/A V6T 1Z3
Canada

Dear Dr. Teves:

Congratulations, your manuscript titled "Dynamic regulation of murine RNA Polymerase III transcription during heat shock stress" is accepted for publication in GENETICS! Many thanks for submitting your research to the journal.

To Proceed to Publication:

1. Format your article according to GENETICS style: <https://academic.oup.com/genetics/pages/general-instructions>
2. Ensure that you comply with data and community resource citation guidelines: <https://academic.oup.com/genetics/pages/general-instructions#Data-Policy>
3. Upload your final files at <https://genetics.msubmit.net>
4. Add oupsupport@scipris.com and genetics.oup@novatechset.com (or the domains @scipris.com and @novatechset.com) to your email program's "safe senders" list. You will be contacted by both at various points during the production process.

Notes:

- Your currently-accepted manuscript (unedited, as submitted, reviewed, and accepted) will be published at GENETICS and deposited into PubMed as an Advance Access article. Notify sourcefiles@thegsajournals.org before signing your license if you do not wish to publish your article via Advance Access.
- We invite you to submit an original color figure related to your paper for consideration as cover art. Please email your submission to the editorial office or upload it with your final files. You can submit a small-sized image for evaluation, and if selected, the final image must be a TIFF file 2513px wide by 3263px high (8.375 by 10.875 inches; resolution of 600ppi). Please avoid graphs and small type.
- After files are sent to Oxford University Press we use SciPris to manage article licensing and payment. If you do not have a SciPris account, you will receive an email from no-reply@scipris.com to sign up to use Oxford University Press' author portal. After logging in, follow the online instructions to sign your license and arrange any payment due.

If you have any questions or encounter any problems while uploading your accepted manuscript files, please email the editorial office at sourcefiles@thegsajournals.org.

Sincerely,

Oliver Rando
Associate Editor
GENETICS

Approved by:
Karen Arndt
Senior Editor
GENETICS

Review comments (if applicable):